# Distinct patterns of microbial association across deep-sea corals from the Western Pacific Magellan Seamounts

Weizhi Song,[1,2,3] Shan Zhang,[1,2,4] Maeva Perez,[5] Jiasui Li,[6] Haiying Ma,[1] Torsten Thomas,[7] Jian-Wen Qiu,[5] Pei-Yuan Qian[1,2,3]

**ABSTRACT**    Ahermatypic corals are common inhabitants in the Magellan Seamounts of the Western Pacific Ocean, yet their microbiomes are largely unexplored. In the present study, we used 16S rRNA gene amplicon sequencing targeting the V4 variable region to characterize the microbiomes of 30 deep-sea coral samples from 9 coral families collected from this area, including members of the families Schizopathidae, Victorgorgiidae, and Chrysogorgiidae, whose microbiomes had not been previously described. Our analyses revealed distinct patterns of microbial association between the coral families, with most coral samples being dominated by single amplicon sequence variants belonging to 11 prokaryotic genera. Ammonia-oxidizing archaea of the genus *Nitrosopumilus* were abundant exclusively in schizopathid corals, with relative 16S rRNA gene read abundances ranging from 29.4% to 99.8%. In contrast, *Nitrosopumilus* was either absent or constituted no more than 5.3% of the reads in the remaining coral families. This may be attributed to the catabolism of the protein-rich zooplankton preferred by schizopathid corals, which could, in turn, facilitate ammonia-driven carbon fixation within the holobiont. Three cladopathid corals hosted abundant sequences of two distantly related bacteria capable of utilizing nitric oxide, which could be used by the symbionts either to generate oxygen for aerobic metabolisms or be reduced as a defense against the host's antibacterial activity. The distinct patterns of microbial association between coral taxa indicate that the microbiomes have differential roles in the adaptation of the hosts to specific ecological niches in the deep-sea environments.

**IMPORTANCE** Microbiomes play crucial roles in host development, physiology, and health, especially in the deep-sea environments. In this study, we collected 30 deep-sea corals from the Western Pacific Magellan Seamounts at depths ranging from 805 to 5,572 m. These samples spanned nine coral families, including three whose microbiomes have not been previously described. Our analyses revealed distinct patterns of microbial association between coral taxa. A majority of the deep-sea corals were dominated by single microbial species, indicating strong selection for certain microbial symbionts and thus functions, such as chemolithoautotrophy, the production of oxygen or secondary metabolites. Furthermore, we observed an overwhelming dominance of sequences from the ammonia-oxidizing archaeal genus *Nitrosopumilus* exclusively in black corals from the family Schizopathidae, a phenomenon not previously reported. This may be attributed to the catabolism of the protein-rich zooplankton preferred by the schizopathid corals, which could, in turn, facilitate ammonia-driven carbon fixation within the coral holobiont.

**KEYWORDS**    deep sea, coral microbiome, Western Pacific Ocean, Magellan Seamounts

Ahermatypic corals are important habitat-forming species in the deep sea and are prevalent in the Western Pacific Ocean, including members of the black coral family Schizopathidae (1) and the octocoral families Primnoidae (2), Coralliidae (3),

Address correspondence to Weizhi Song, songwz03@gmail.com, or Pei-Yuan Qian, boqianpy@ust.hk.

Weizhi Song and Shan Zhang contributed equally to this article. The author order was determined alphabetically.

The authors declare no conflict of interest.

Chrysogorgiidae (4, 5), Victorgorgiidae (6), and Keratoisididae (7). The black corals (subclass Hexacorallia, order Antipatharia) and octocorals (subclass Octocorallia) have different structural or nutritional properties; for example, the skeletons of black corals are primarily composed of chitin and protein (8), whereas those of octocorals contain abundant carbonates (or their polymorphs), gorgonin, or collagen (9). In addition, black corals primarily feed on zooplankton (8) and have only been recently reported to take up phytoplankton by *Antipathella wollastoni* (10), whereas octocoral species have been widely reported to consume phytoplankton (11, 12), suggesting that black corals may rely more exclusively on zooplankton than octocorals. Differences in physiology and diet might also occur down to lower taxonomic levels. For example, the shallow-water black corals *Antipathes atlantica* and *Stichopathes luetkeni* exhibit reliance on different components of the planktonic community, with *A. atlantica* consuming more micro- and mesozooplankton compared to *S. luetkeni* (13). In addition, isotopic analysis revealed that the deep-sea black coral *Bathypathes arctica* occupies a higher trophic position within the food web than the deep-sea octocorals *Paragorgia arborea* (family Coralliidae) and *Primnoa resedaeformis* (family Primnoidae) (14), while the deep-sea black coral *Leiopathes glaberrima* (family Leiopathidae) had a lower trophic position compared to some octocorals (15). This shows that different coral taxa occupy distinct ecological and nutritional niches.

Corals form complex associations with microorganisms, including bacteria (16), archaea (17), and unicellular eukaryotes (18), which support host health, immunity, metabolism, and environmental adaptation (19). Microbial associations of corals are likely affected by both the traits of corals (e.g., diet [20], morphology [21], and ecology [22]) and environmental factors (e.g., habitat depth [23] and light availability [24]). Deep-sea coral-associated microorganisms are considered particularly important for host metabolism because of the extreme conditions in the deep sea, such as the lack of light-dependent symbionts that provide food and energy to the host (24). However, the microbiomes of deep-sea corals have been characterized primarily on a limited number of octocoral (23, 25–32) and scleractinian coral species (24, 27, 29, 31, 33, 34). These studies have examined the influence of environmental factors, such as depth and temperature, on microbiome composition (23, 33), the adaptations of symbiotic microorganisms to a coral-associated lifestyle in the deep sea (24, 26, 27), and the potential contributions of microbial symbionts to the survival of coral hosts under deep-sea conditions (28, 31). Some studies have also reported differences in microbiome composition among deep-sea coral species (24, 29, 30, 33); however, the underlying mechanisms driving these differences in the deep-sea environments have rarely been explored. It is worth noting that only two studies have investigated the microbiomes associated with deep-sea black corals. One investigated the response of bacterial symbionts to oil exposure in *Leiopathes glaberrima* and *Sibopathes* spp. corals (35). The other examined the bacterial community of an unidentified black coral from a Gulf of Alaska seamount using Sanger sequencing of 16S rRNA gene clone libraries, revealing that Alphaproteobacteria, Bacteroidetes, and Firmicutes were the three most dominant bacterial groups (36).

Here, we hypothesized that deep-sea corals from different taxonomic groups have distinct microbiomes due to differences in hosts' physiology, diet preferences, and ecology. To test our hypothesis and to fill the knowledge gaps on yet-undescribed microbiomes of many deep-sea corals, especially the severe sample bias on the microbiome of deep-sea black corals, we collected 30 deep-sea coral samples spanning 9 families from the Magellan Seamounts in the Western Pacific Ocean at depths ranging from 805 to 5,572 m (File S1). Specifically, three of these nine families have no previously described microbiomes, including one black coral family Schizopathidae and two octocoral families Victorgorgiidae and Chrysogorgiidae (Fig. S1). We investigated the deep-sea coral microbiomes with surrounding seawater and sediment samples by performing amplicon sequencing of the V4 region of the 16S rRNA gene and compared their differences in microbial associations.

## MATERIALS AND METHODS

### Sample collection and processing

Coral specimens (Fig. 1) and bulk environmental samples, including seawater and sediment, were collected at depths ranging from 805 to 5,572 m across six seamount guyots and the adjacent Pigafetta Basin in the Western Pacific Ocean (12°13′–23°54′N, 148°34′–156°32′E, Fig. 2A; File S1). Samples were collected during a series of dives conducted with the Jiaolong human-operated vehicle (HOV) between August 18th and September 11th in 2024. Coral samples were retrieved using the HOV mechanical arms and placed in lidded biological sample boxes designed to minimize seawater exchange during ascent (Fig. 2B and C). Upon retrieval, faunal specimens were photographed on deck and immediately transferred to a biological safety cabinet (BSC) in the onboard laboratory. Since multiple specimens from the same dive were stored in the same biological sample box, coral tissues were rinsed three times with sterilized seawater (pre-filtered through 0.22 µm pore-size membrane filters [MCE, Millipore]) in the BSC to remove loosely associated microorganisms that may have been introduced during the sampling process, such as from physical contact between different individuals and/or attachment to sample trays during photography. The tissues were subsequently

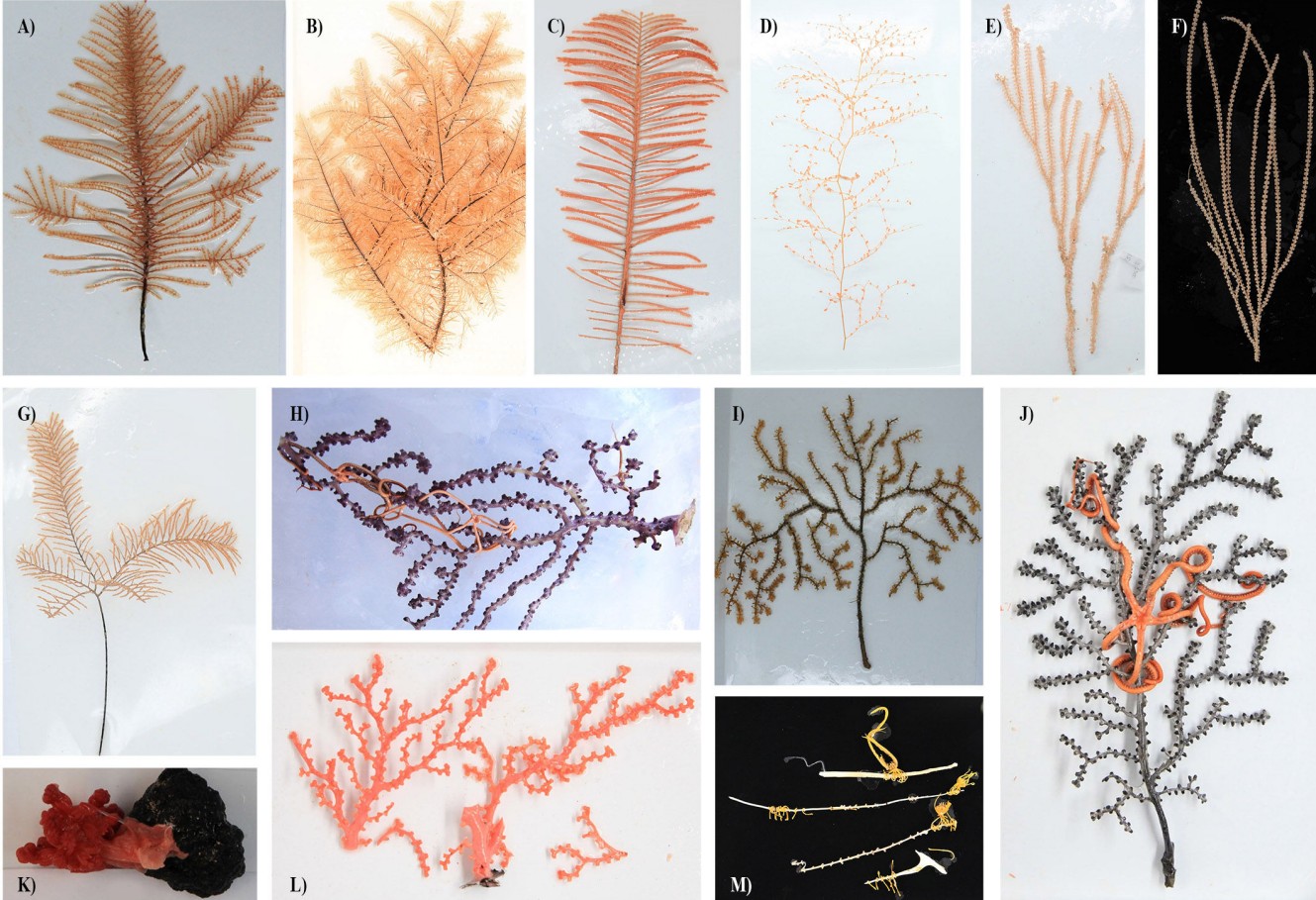

FIG 1   Representative coral specimens with taxonomy based on the 28S rRNA and/or COI genes. (A) Family Cladopathidae (JL304_B13); (B) Family Schizopathidae, genus *Parantipathes* (JL307_B11); (C) Family Schizopathidae, genus *Bathypathes* (JL311_B01); (D) Family Chrysogorgiidae, genus *Chrysogorgia* (JL313_B01); (E) Family Primnoidae, genus *Calyptrophora* (JL300_B21); (F) Family Primnoidae, genus *Narella* (JL317_B09); (G) Family Schizopathidae (JL317_B01); (H) Family Victorgorgiidae, genus *Victorgorgia* (JL300_B04); (I) Family Paramuriceidae, genus *Acanthogorgia* (JL312_B19); (J) Family Paramuriceidae, genus *Paramuricea* (JL303_B26); (K) Family Coralliidae, genus *Anthomastus* (JL306_B04); (L) Family Coralliidae, genus *Hemicorallium* (JL303_B27); and (M) Family Keratoisididae, genus *Keratoisis* (JL316_B01).

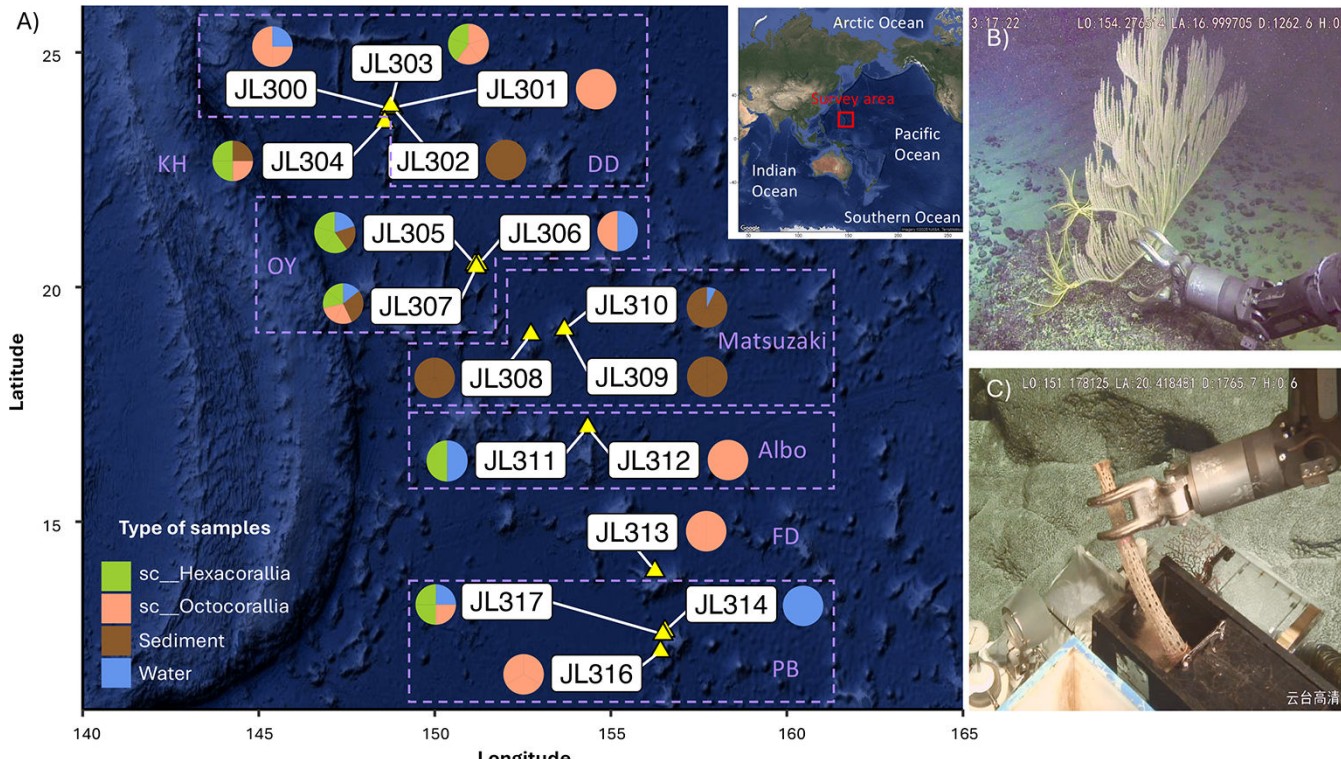

**FIG 2** The location of dive stations and survey approaches. (A) The map showed dives conducted in the highlighted survey area (red square on the global map). The station map was created using the R package ggmap v4.0.1, which embeds the geographic map by accessing Google Maps. The pie charts illustrate the types of samples collected during each dive, with dashed lines highlighting dives from the same seamount. Sample collection methods included (B) the use of mechanical arms and (C) lidded biological sample boxes aboard the Jiaolong HOV. DD, Digital-Depth Guyot; KH, Ko-Hakucho Guyot; OY, O-Yatagarasu Guyot; FD, Fedorov Guyot; PB, Pigafetta Basin.

transferred into sterile microcentrifuge tubes, either snap-frozen in liquid nitrogen and stored at −80°C or preserved in absolute ethanol at 4°C. Seawater and sediment samples were collected using a Niskin water sampler and push corers mounted on the HOV, respectively. For each seawater sample, 1.5–2 L were filtered through 0.22 μm membrane filters, and the filters were preserved in cryogenic storage vials and stored at −80°C. Sediments were subsampled into sterile tubes and stored at −80°C until further processing.

## DNA extraction, library construction, and sequencing

Genomic DNA was extracted using the CTAB Protocol by NovoGene Co. Ltd (Tianjin, China). DNA quality was assessed by agarose gel electrophoresis, Qubit, and the Agilent 5400 (Agilent, CA, USA). For genome skimming (low-depth metagenomic sequencing), genomic DNA was sheared into ~350 bp fragments using a Covaris Ultrasonicator (Covaris, Massachusetts, USA). For amplicon sequencing, the V4 region of the 16S rRNA gene was amplified using the primers 515F (5′-GTGCCAGCMGCCGCGGTAA-3′) and 806R (5′-GGACTACHVGGGTWTCTAAT-3′) (37). Sterilized water was used as a negative control in PCR amplification. Sequencing libraries were prepared using the NEB Next Ultra II FS DNA PCR-free Library Prep Kit (New England Biolabs, USA, Catalog E7430L) following the manufacturer's recommendations and sequenced on Illumina NovaSeq X Plus and NovaSeq 6000 platforms with PE150 strategy (Illumina, San Diego, CA, USA) (File S1; sample metadata.txt).

## Host classification and phylogeny inference

Genome skimming data for each coral sample were trimmed using Trimmomatic v0.39 (20) (settings: CROP:145 HEADCROP:5 LEADING:20 TRAILING:20 SLIDINGWINDOW:4:25 MINLEN:35) and assembled using SPAdes (21) (File S1; sample metadata.txt). Corals were classified either by Sanger sequencing of the 28S rRNA gene with primer pairs C2 (5′-GAAAAGAACTTTGRARAGAGAGT-3′) and D2 (5′-TCCGTGTTTCAAGACGGG-3′) (38), and/or the cytochrome c oxidase subunit I (COI) gene with primer pairs dgLCO1490 (5′-GGTCAACAAATCATAAAGAYATYGG-3′) and dgLCO2198 (5′-TAAACTTCAGGGTGACCAA ARAAYCA-3′) (39), or by identification of the two marker genes from assemblies of the genome skimming data of the coral host. Specifically, barcoding sequences were retrieved from genome-skimming assemblies by performing BLASTN (40) searches between the metagenomic assemblies and the barcoding sequences of other coral samples obtained via the above Sanger sequencing, or against reference sequences of the Sanger-sequenced corals retrieved from the NCBI nt database. Reference sequences were selected by retaining the top five best hits from the NCBI nt database (41) for each marker sequence. Sequences were aligned using Mafft v7.490 (42) (default setting) and trimmed with Gblocks v0.91b (43) (settings: -t = d -b3 = 24 -b4 = 6 -b5 = a). Phylogenetic trees were inferred using IQ-TREE v2.2.0 (44) under model "GTR + I + G" and visualized with iTOL (45).

## Amplicon sequencing data processing

Paired-end reads were merged using FLASH v1.2.11 (46) (settings: g -q 19 u 15 n 5 L 15 --overlap_diff_limit 5 --overlap_diff_percent_limit 20) and quality filtered using fastp v0.23.1 (47) (settings: -q 19 u 15). Sequencing reads were denoised (error-corrected) using the UNOISE3 algorithm (48) (default setting) implemented in Usearch v11.0.667 (49). Chimeric sequences were removed using the uchime2_ref module (50) (settings: -strand plus -mode high_confidence) in referencing mode with the SILVA database (v138.2) (51). Eukaryotic single amplicon sequence variants (ASVs) were identified and removed by searching against the NCBI non-redundant nucleotide (nt) database using BLASTN (40). An ASV was ignored if more than 10% of its top 10 best hits are from Eukaryotes. Taxonomic classification was performed using BLCA (52) against the GTDB database (release 220) (53). ASVs that could not be classified by BLCA were further identified by performing BLASTN searches (default settings) against the nt database and assigned as bacterial or archaeal if all top 20 hits matched the respective domain.

## Statistical analysis

Permutational Multivariate Analysis of Variance (PERMANOVA) was performed using the "adonis2" function from the vegan v2.7-1 R package. Non-metric multidimensional scaling (NMDS) was conducted with the "metaMDS" function from the same R package. Both analyses were based on microbial taxonomy merged at the genus level, with the Bray-Curtis distance metric being used to calculate the distances. Water and sediment samples were excluded from PERMANOVA.

## RESULTS AND DISCUSSION

Our results revealed that microbial community structures (ASV merged at the genus level) were distinct between the black corals and the octocorals (PERMANOVA, $R^2$ = 0.18, *P* value = 0.001), as well as across different coral families (PERMANOVA, $R^2$ = 0.51, *P* value = 0.001) and different genera (PERMANOVA, $R^2$ = 0.77, *P* value = 0.001) (Fig. 3). Archaea accounted for 29.4%–99.8% of the sequencing reads in the nine black coral samples from the family Schizopathidae (Fig. 4). In contrast, they were absent in the remaining 21 coral samples, except for two corals from the families Cladopathidae and Chrysogorgiidae, where archaeal reads accounted for less than 5.3% of all 16S rRNA gene reads. Further analysis revealed that ASV1 from the genus *Nitrosopumilus* dominates in seven archaea-abundant schizopathid corals (five of which belong to

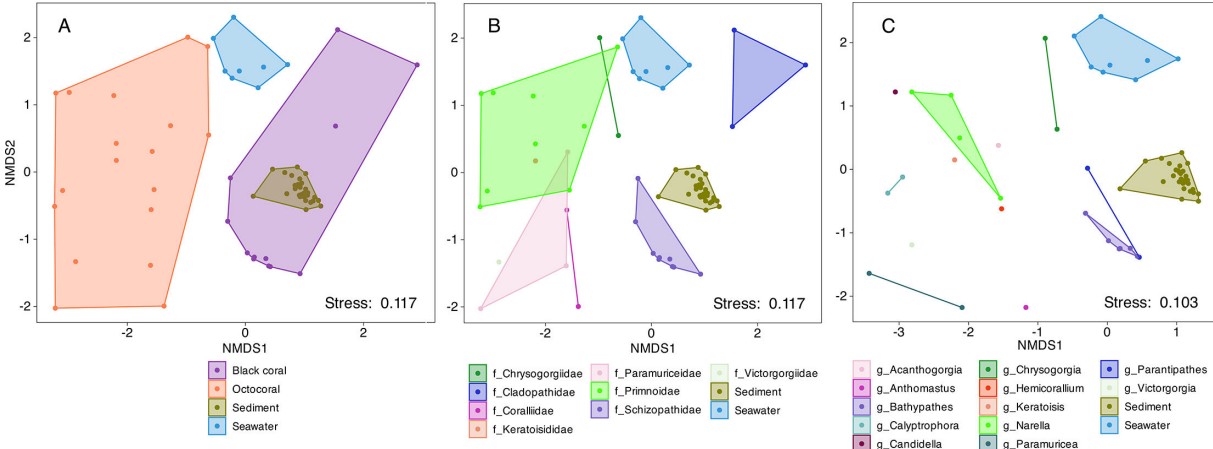

**FIG 3** NMDS plot showing the similarity of microbial communities (at genus level): (A) between the black corals and the octocorals, (B) among coral families, and (C) among coral genera, as well as compared to seawater and sediment samples. Coral samples that could not be assigned to a specific genus were excluded from the genus-level NMDS analysis. Coral samples from the same group were enclosed with polygons.

the genus *Bathypathes*), with relative read abundance ranging from 78.3% to 100% of all archaeal reads (Table 1). Another two *Nitrosopumilus* ASVs (i.e., ASV28 and ASV27) dominate the archaeal sequences in the remaining two *Bathypathes* (JL305_B10 and JL311_B01), whose 28S rRNA genes share more than 99.5% similarity with the above five *Bathypathes* corals (Fig. 5; Table 1). ASV27 and ASV28 differ by one base pair and share 97.2% and 97.6% similarity with ASV1, respectively, suggesting that they may belong to two different *Nitrosopumilus* species. *Nitrosopumilus*, a genus of ammonia-oxidizing archaea (AOA) that are ubiquitous in marine and terrestrial environments (54), has been reported as a dominant member in some deep-sea sponges and has the genetic capacity for primary production within sponge holobionts (55, 56). Although members of the genus *Nitrosopumilus* have been commonly found in shallow-water corals, they have never been reported as a dominant taxon (57–62), including in the deep-water corals (33). This may be due to the previous use of primer pairs that have limited efficacy in amplifying archaeal sequences (23–25, 33). The apparent high abundance and prevalence of *Nitrosopumilus* in the deep-sea schizopathid corals (Fig. 5) may be attributed to the hosts' dietary preferences. Specifically, the digestion and catabolism of protein-rich zooplankton releases ammonia (63), which could, in turn, support the growth of *Nitrosopumilus* and its carbon fixation, as has been postulated based on metagenomic analysis for others deep-sea corals (27, 64) and sponges (55).

In addition to the absence of *Nitrosopumilus* sequences, the three black corals from the family Cladopathidae (with no less than 98.3% similarities of the 28S rRNA genes) mainly contained sequences from the bacterial placeholder genera UBA11136 (class Alphaproteobacteria) and UBA2767 (family Methyloligellaceae) (Fig. 5). UBA11136 has been reported to exhibit high transcription levels of the *nod* gene, which has been proposed to dismutate nitric oxide (NO) into dinitrogen and oxygen in marine oxygen-deficient zones (65). UBA2767 has been reported to have the genetic capacity to oxidize NO into nitrate in sponges using a NO dioxygenase, an enzyme implicated in detoxification (66, 67). Shallow-water corals, like many other metazoans, produce NO as a signaling molecule to control processes such as apoptosis (68), or to be used in antibacterial defense (69). Therefore, the deep-sea corals likely produce NO, which might be used by UBA11136 to generate oxygen for aerobic processes or be reduced to protect UBA2767 against its antibacterial effect.

Four octocorals from the family Primnoidae (including one from the genus *Candidella* and three from the genus *Narella*) contained abundant sequences from the bacterial placeholder genus CAKMZU01 (class Gammaproteobacteria, order Pseudomonadales), with relative abundances ranging from 44.5% to 95.4% (Fig. 5). A member of the

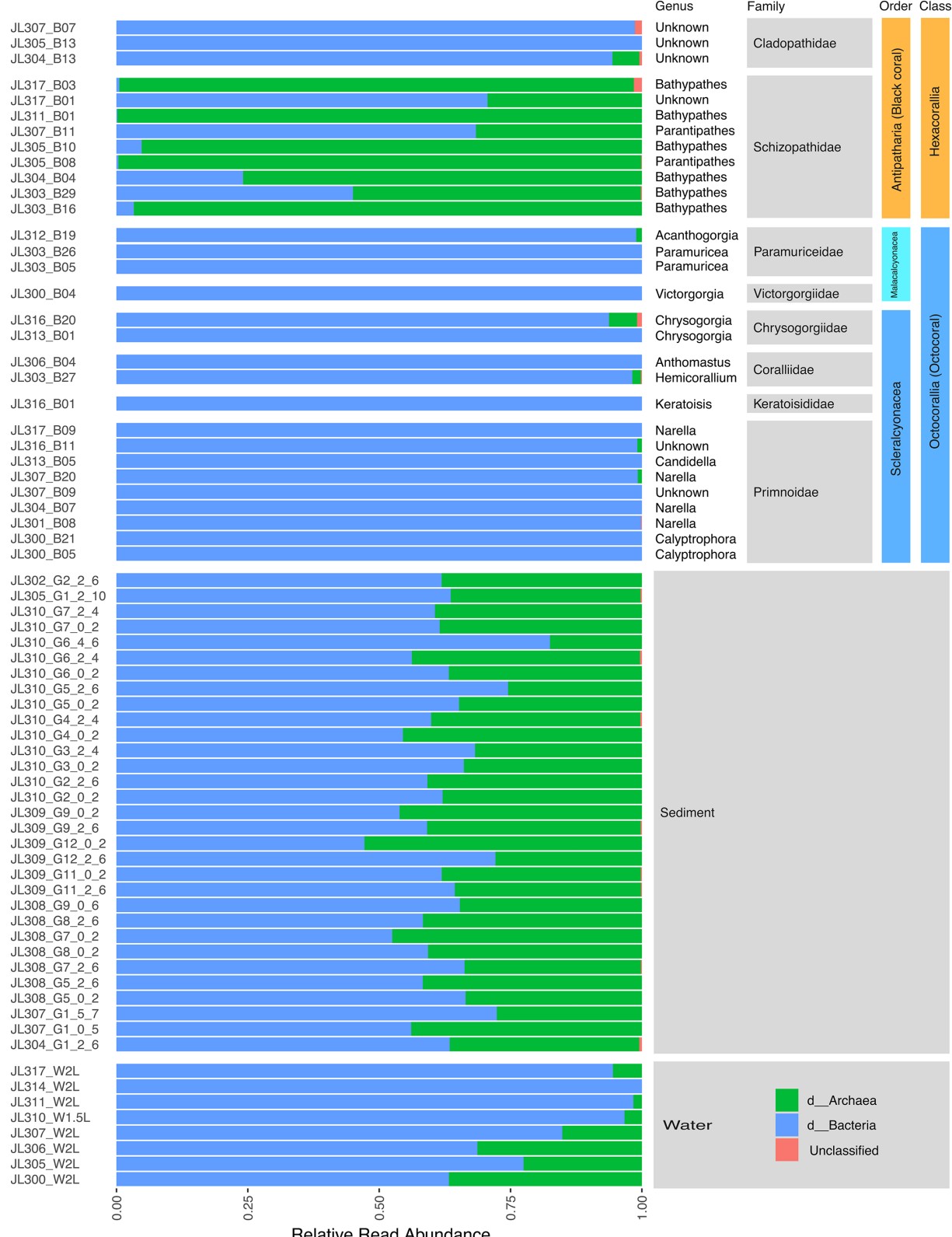

**FIG 4** Community structure of the microbiomes across coral, seawater, and sediment samples. ASVs with less than 0.1% relative read abundance were excluded.

genus CAKMZU01 (GCA_929200485.1) has also been previously found associated with a shallow-water octocoral and had abundant biosynthetic gene clusters for a variety of

**TABLE 1** Relative read abundance of the three *Nitrosopumilus* ASVs as a proportion of all archaeal reads within the microbial communities of the nine schizopathid corals

| Coral genus | Sample ID | Relative read abundance (%) | | |
|---|---|---|---|---|
| | | ASV1 | ASV27 | ASV28 |
| *Bathypathes* | JL303_B16 | 99.99 | 0 | 0 |
| | JL303_B29 | 99.97 | 0 | 0 |
| | JL304_B04 | 99.48 | 0 | 0 |
| | JL317_B03 | 84.04 | 0 | 0 |
| | JL311_B01 | 0 | 99.89 | 0.04 |
| | JL305_B10 | 0.04 | 0.05 | 99.81 |
| *Parantipathes* | JL305_B08 | 81.7 | 0 | 0 |
| | JL307_B11 | 78.17 | 0 | 0 |
| Unknown | JL317_B01 | 97.49 | 0 | 0 |

secondary metabolites, including all eight B vitamins (70). The deep-sea representatives of the genus CAKMZU01 may thus also possess the capabilities to biosynthesize a range of secondary metabolites that could benefit the coral host in the deep sea. Another two octocorals from the genus *Calyptrophora* within the family Primnoidae (Fig. S4) contain abundant bacterial reads from the genus JABDGQ01 (class Chlamydia, order Chlamydiales), with relative read abundances of 59.5% and 24.9%, respectively (Fig. 5). Chlamydiales are common obligate intracellular pathogens of animals and humans, and while they have been found in shallow-water corals, they remain poorly characterized (71–73). The abundance of Chlamydiales in the two corals from the genus *Calyptrophora* may suggest potential bacterial infections; however, this will require further work to confirm.

One octocoral from the genus *Chrysogorgia* (family Chrysogorgiidae) was dominated by the genus *Ruegeria* (relative read abundance 58.7%) (Fig. 5), which has been proposed as a potential probiotic to protect corals from bleaching (74). However, *Ruegeria* was absent from the other *Chrysogorgia* coral, indicating that the genus might not be

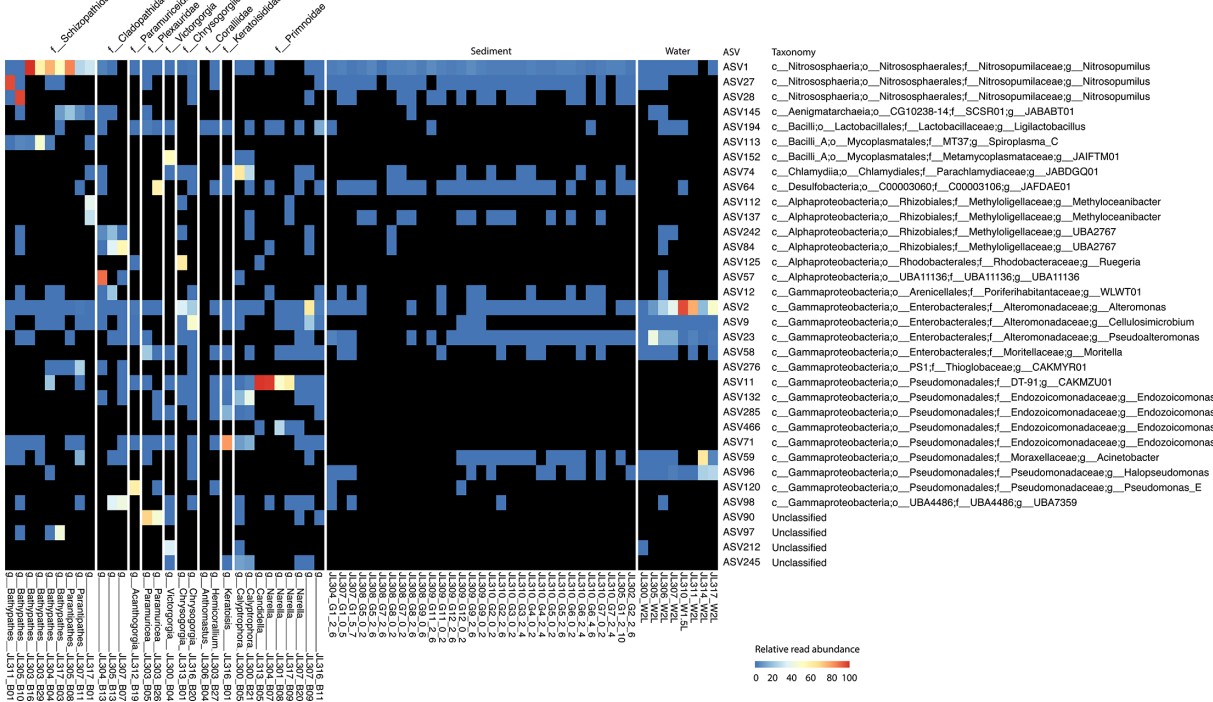

**FIG 5** ASVs with ≥10% relative read abundance in at least one sample across coral, seawater, and sediment samples. Black boxes indicate that ASVs were absent in the samples.

essential for the host. Additionally, sequence reads belonging to the genus *Alteromonas*, which is also prevalent in the surrounding seawater, were abundant in the two *Chrysogorgia* corals, with relative read abundances of 31.3% and 21.0%, respectively (Fig. 5). Members of the genus *Alteromonas* have been found to be involved in carbon cycling in marine oxygen minimum zones (75), and many of them play a role in dimethyl-sulfonioproprionate metabolism and nutrient cycling (16), and could thus support the coral's metabolic homeostasis (76).

Octocorals from the remaining families, excluding the two Coralliid corals, were often dominated by one ASV. For instance, corals from the genera *Victorgorgia* (family Victorgorgiidae) and *Acanthogorgia* (family Paramuriceidae), as well as the family Keratoisididae, were dominated by sequences from the bacterial genera JAIFTM01, *Pseudomonas_E,* and *Endozoicomonas*, respectively (Fig. 5). *Endozoicomonas* has been found to dominate the microbiome of some shallow-water corals and was initially thought to be restricted to such habitats (73, 77, 78) before being also found in deep-sea corals (23, 79, 80). Our observations further confirm the presence of *Endozoicomonas* in deep-sea corals, and sample JL316_B01 (from the genus *Keratoisis*) represents the greatest depth (1,697 m) and the coldest water (~2.5°C), where coral-associated *Endozoicomonas* has been detected. It is worth mentioning that, in addition to the dominance of *Endozoicomonas* (bacterial class Gammaproteobacteria, 77.23% relative read abundance) observed in our bamboo coral sample JL316_B01 (family Keratoisididae, genus *Keratoisis*), bamboo corals collected from the Gulf of Alaska seamount were reported to harbor abundant Alphaproteobacteria, Firmicutes, Bacteroidetes, and Acidobacteria (36). It also should be noted that the coral taxa mentioned above are represented by only one or two samples, which is insufficient to capture the full spectrum of microbial associations or to account for intra-family or intra-genus variation in microbial associations.

In conclusion, distinct patterns of microbial association were observed between different coral taxa. A majority of the deep-sea coral samples (19 out of 30) were dominated by single ASVs belonging to one archaeal and ten bacterial genera. In contrast, only a limited number of microbial taxa, such as *Endozoicomonas* (81, 82), Spirochaetes (83, 84), and the photosynthetic *Prosthecochloris* (85), have been identified as dominant species in shallow-water corals. The frequently observed dominance of single ASVs in these deep-sea corals may indicate strong selection for certain microbial symbionts and thus functions, such as chemolithoautotrophy or the production of oxygen or secondary metabolites. The distinct patterns of symbiont enrichment between coral taxa indicate that the microbiome has differential roles in adapting the host to specific ecological niches in the deep-sea environments. Future studies on whole-genome metagenomic and metabolomic analyses could provide further insight into the specific functional roles of these symbionts, shedding light on how they contribute to the adaptation and survival of these rarely studied deep-sea corals in the deep sea.

## ACKNOWLEDGMENTS

We thank the crew of R/V Shen-Hai-Yi-Hao, the operation team of the HOV Jiaolong, and on-board scientists during the 2024 International Cruise of Digital DEPTH (https://digitaldepth.ndsc.org.cn) for collecting the deep-sea corals. We especially thank Meiling Ge, Ruiyan Zhang, and Erika Gress for their contributions to the morphology-based coral classifications.

This work was supported by the grants from the Southern Marine Science and Engineering Guangdong Laboratory (Guangzhou) (2021HJ01, HJRC2022001, and SMSEGL24SC01), the Hong Kong Special Administrative Region government (16103925, 16101822, C2013-22GF), and the Otto Poon Center for Climate Resilience and Sustainability at the Hong Kong University of Science and Technology (CCRS25SC01).

## AUTHOR AFFILIATIONS

[1]Southern Marine Science and Engineering Guangdong Laboratory (Guangzhou), Guangzhou, China

[2]Department of Ocean Science, The Hong Kong University of Science and Technology, Hong Kong, China

[3]Otto Poon Center for Climate Resilience and Sustainability, The Hong Kong University of Science and Technology, Hong Kong, China

[4]Department of Pharmacology and Pharmacy, LKS Faculty of Medicine, The University of Hong Kong, Hong Kong, China

[5]Department of Biology, Hong Kong Baptist University, Hong Kong SAR, China

[6]School of Life and Environmental Sciences, The University of Sydney, Sydney, Australia

[7]Centre for Marine Science and Innovation, School of Biological, Earth and Environmental Sciences, University of New South Wales, Sydney, New South Wales, Australia

## AUTHOR ORCIDs

Weizhi Song  http://orcid.org/0000-0001-5890-5361
Shan Zhang  http://orcid.org/0000-0002-1246-7017
Maeva Perez  http://orcid.org/0000-0003-4532-8299
Jiasui Li  http://orcid.org/0000-0001-8452-3674
Haiying Ma  http://orcid.org/0000-0002-5969-900X
Torsten Thomas  http://orcid.org/0000-0001-9557-3001
Jian-Wen Qiu  http://orcid.org/0000-0002-1541-9627
Pei-Yuan Qian  http://orcid.org/0000-0003-4074-9078

## AUTHOR CONTRIBUTIONS

Weizhi Song, Conceptualization, Formal analysis, Investigation, Methodology, Visualization, Writing – original draft, Writing – review and editing, Validation | Shan Zhang, Conceptualization, Data curation, Formal analysis, Investigation, Methodology, Visualization, Writing – original draft, Writing – review and editing | Maeva Perez, Formal analysis, Investigation, Methodology | Jiasui Li, Formal analysis, Methodology, Visualization | Haiying Ma, Formal analysis, Investigation | Torsten Thomas, Formal analysis, Investigation, Methodology, Visualization, Conceptualization, Writing – review and editing | Jian-Wen Qiu, Investigation, Resources, Writing – review and editing | Pei-Yuan Qian, Funding acquisition, Project administration, Resources, Supervision, Writing – review and editing

## DATA AVAILABILITY

Metadata for the coral, seawater, and sediment samples are available in File S1; sample metadata.txt. The genome skimming and amplicon sequencing data have been deposited under the NCBI BioProject PRJNA1261858, with accession numbers for individual samples provided in File S1. The 28S rRNA and cytochrome c oxidase subunit I (COI) gene sequences of the corals are provided in Files S2 and S3, respectively.

## ADDITIONAL FILES

The following material is available online.

### Supplemental Material

**Supplemental figures (Spectrum02093-25-s0001.docx).** Figures S1 to S4.
**Supplemental File S1 (Spectrum02093-25-s0002.txt).** Metadata of the coral samples.
**Supplemental File S2 (Spectrum02093-25-s0003.txt).** The 28S rRNA gene sequences of the coral samples.

**Supplemental File S3 (Spectrum02093-25-s0004.txt).** The Cytochrome c oxidase subunit I (COI) gene sequences of the coral samples.

Open Peer Review

**PEER REVIEW HISTORY (review-history.pdf).** An accounting of the reviewer comments and feedback.

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
