## [Reviewer comments · Microbiology Spectrum]

Microbiology Spectrum

Distinct Patterns of Microbial Association Between Deep-sea Corals from the Western Pacific Magellan Seamounts

Weizhi Song, Shan Zhang, Maeva Perez, Jiasui Li, Haiying Ma, Torsten Thomas, Jian-Wen Qiu, and Pei-Yuan Qian

Corresponding Author(s): Pei-Yuan Qian, Hong Kong University of Science and Technology

Review Timeline:

Submission Date:	July 9, 2025
Editorial Decision:	August 12, 2025
Revision Received:	September 8, 2025
Accepted:	October 10, 2025

Editor: Konstantinos Kormas

Reviewer(s): The reviewers have opted to remain anonymous.

Transaction Report:

DOI: <https://doi.org/10.1128/spectrum.02093-25>

Re: Spectrum02093-25 (**Distinct Patterns of Microbial Association Between Deep-sea Corals from the Western Pacific Magellan Seamounts**)

Dear Prof. Pei-Yuan Qian:

Thank you for the privilege of reviewing your work. Below you will find my comments, instructions from the Spectrum editorial office, and the reviewer comments.

Revision Guidelines

Sincerely,
Konstantinos Kormas
Editor
Microbiology Spectrum

Reviewer #1 (Comments for the Author):

The authors conduct a 16S metabarcoding survey of several deep-sea corals, water, and sediment from the Magellan Seamounts in the western pacific to characterize those corals' microbiomes. This study focuses on under-studied coral taxa including black corals and several octocoral families and identifies *Nitrosopumilus* as an abundant member of several black coral microbiomes and detects general differences between coral taxa at multiple taxonomic levels. The methods seem appropriate. I

suggest that the paper be accepted with minor revisions.

My main criticism is a lack of important details in the methods which I list below and the interpretation of the results. The authors propose that black corals associate with *Nitrosopumilus* because zooplankton comprise a larger proportion of their diet compared to other coral taxa. The literature cited does not support the claim that black corals rely on more zooplankton. Several of the cited works do not even refer to the topic of the sentences they are associated with. I don't know if a citation tool messed up or the works were not thoroughly read. The authors also imply that generally octocorals eat primarily phytoplankton which is not true. In general, I think more uncertainty should be expressed. Something like black corals MAY rely on less phytoplankton and this MAY be the basis of their association with *Nitrosopumilus*. Otherwise, the rest of the discussion was good and the authors handled the other abundant microbes well.

Line 28: remineralization of zooplankton? There must be a more direct way to communicate this. You mean the nitrogen waste. Is there evidence that schizopathids prefer zooplankton more so than the other deep-sea corals?

Line 30: cladopathid corals "hosted" ...

Line 39: Microbiomes

Line 50: disagree with likely. Should say MAY be attributed to

Line 67: It is not accurate to generalize octocorals as only consuming phytoplankton.

Ref 11 shows that some shallow water octos in the red sea eat exclusively phytoplankton. This is not very informative for deep-sea corals where there isn't light

Ref 12 doesn't even mention octocorals at all.

Ref 13 shows *P. clavata* can capture things other than zooplankton.

There are several studies on deep-sea corals diets. I think you should reframe this by saying black corals may rely more exclusively on zooplankton than octocorals.

The stable isotope results in Ref 16 suggest that *Paramuricea macrospina* primarily eats zooplankton and that *Leiopathes* has a lower contribution of zooplankton in its diet. This runs counter to the generalizations proposed by the authors.

Line 96: Missing citation from Dannenberg. This was published as Vohsen et al 2020. Microbiome. Deep-sea corals provide new insight into the ecology, evolution, and the role of plastids in widespread apicomplexan symbionts of anthozoans.

Line 162: What were the parameters used for all these tools?

Line 167: you mean if more than 1 of the top 10 hits are eukaryotes? This is an unusual criterion. Why was this chosen?

Line 186: were the permanova analyses conducted on ASVs or higher taxonomic level? What distance metric was used? Water and sediment should have been excluded from order, family, and genus comparisons. Is this how they were conducted?

All details regarding the stats analyses are missing from the methods

Table 1 should state the species for each of those samples

Line 212: Ref 29 does not conclude that *Nitrosopumilus*' chemoautotrophy occurs in other deep-sea corals. It discusses chemoautotrophy by sulfur oxidizing SUP05 based on 16S studies and not metagenomics. Reword this to be clear.

Line 223: Were the permanova analyses also conducted on genus level? What distance metric was used for NMDS and permanova? Do you mean ASVs that could not be assigned to genus were excluded?

Line 225: that figure should also give the lowest taxonomic id for each sample

Line 196: B01 does not say *Bathypathes* in the figure below

Paramuricea is in the *Paramuriceidae* and not the *Plexauridae*

Figure 5: those "genus" classifications are often just the closest match in GTDBTK database. Provide a higher taxonomic level so they are interpretable

No mention of genome skims in methods. No mention how they got coral barcodes from skims. No mention of genome skims in data availability or are they some of those SRAs?

Was there anything about the bathypathes that could explain why 2 host different *Nitrosopumilus*? Maybe those bathys were from different sites or different depths. Authors should mention that *Nitrosopumilus* is a very common member of the bacterioplankton.

Reviewer #2 (Comments for the Author):

This study uses 16S amplicon surveys of the V4-variable region to examine the bacterial and archaeal communities of 30 deep-sea coral samples, belonging to 9 different coral families. The main limitation of this study is that the specimen collections contain few biological replicates, with many genera or even families at $n=1$ or $n=2$. The authors have done their best to be able to say something meaningful by combining the data at the family level. The results reflect several expected findings plus some bits of new information. For example, it is in no way unexpected based on existing literature to find that the microbial families are different between black corals and octocorals, different between coral families, and different between coral genera. It is similarly not unexpected to find *Endozoicomonas* present in some of the deep-sea coral-associated microbial communities. What is new information is the high relative abundance of archaea, specifically *Nitrosopumilus* in the microbiomes of the black coral family *Schizopathidae*, but not in family *Clathropathidae* nor any of the octocoral families. The study also provides new information in the form of specific ASVs that are dominant in coral families and genera that have not yet been characterized, but with sample

sizes of 1, 2, or even 4, it is of less value since it is unclear how generalizable these data might be to other geographic locations or even other coral individuals. The introduction and results/discussion sections contain appropriate references and place the study in proper context of the existing literature.

Specific comments:

(Note that the line numbers are slightly different between the Word docx version and the PDF version; these numbers refer to the docx version of the manuscript)

Abstract:

- Please specify in the abstract the variable region targeted for 16S amplicon surveys. E.g., "In the present study, we used 16S rRNA gene amplicon sequencing targeting the V4 variable region to characterize the microbiomes..."

Introduction:

- Line 67: The sentence cites [11-13] about octocorals, but reference 12 is about black coral *Antipathella wollastoni*.
- Lines 86-88: Please consider including Gray et al. (<https://academic.oup.com/femsec/article/76/1/109/520994>) for octocorals and Chapron et al. (<https://www.frontiersin.org/journals/microbiology/articles/10.3389/fmicb.2020.00275/full>) for scleractinians.
- Lines 95-97: While it is absolutely true that deep-sea black coral microbiomes are extremely understudied, this sentence could be more accurate. Citation [34] references a master's thesis but there is also one peer-reviewed example (Penn et al., <https://journals.asm.org/doi/full/10.1128/aem.72.2.1680-1683.2006>) albeit not a very helpful one since the scientific name of the black coral sampled is not provided and it from early days of clone libraries. However, the coral is from a seamount in the Gulf of Alaska, so worth making the connection.

Methods:

- Lines 118-120: Please specify whether each coral was in a separate compartment of the collection box, or whether multiple specimens were in the same container (in physical contact with each other). It allows readers to gauge potential for cross-contamination.
- Lines 120-121: How were corals handled during photography? E.g., were sterile forceps/gloved hands used to transfer them from the submersible biobox to sterile trays? Otherwise, be clear on this possible route for contamination.
- Lines 121-123: Specimens rinsed 3x with sterile filtered seawater-good, this definitely helps reduce contact-contamination!
- Line 125: Why are there 2 preservation methods (liquid nitrogen and ethanol)? Were all corals preserved both ways for safety, or were some preserved one way and some the other? Preservation method can affect the diversity detected in the microbiome, so if the sequenced samples are of mixed preservation provenance, you should indicate which samples were preserved which ways, since it could be the cause of some of the patterns you are seeing.
- Figure 1 legend: Panel M is listed between G and H, when it should be at the end.
- Lines 152-153: Please include citations for primers. E.g., the citation for 515F/806R is Caporaso et al. 2011 <https://www.pnas.org/doi/abs/10.1073/pnas.1000080107>.
- Lines 150-156: Please describe the methods used to assess DNA quality; e.g., spectrophotometrically like NanoDrop, fluorometrically like Qubit.
- Lines 150-156: Please describe any positive controls (e.g., mock community) and negative controls (e.g., kit extraction blanks, PCR negatives, sequencing no-template-controls) that were used.
- Lines 173-177: Please provide reference(s) for the primers listed.
- Good job describing the bioinformatics, including citations for tools and version numbers for databases. Noting that the raw data have been made available in NCBI and the supplemental files.

Results and Discussion:

- Table 1: It would be most helpful to include genera names, when possible, for samples in addition to the sample ID numbers.
- Line 198: Given that ASV27 and ASV28 differ by one base pair, they are likely the same organism and that is a sequencing artifact. I agree that ASV1 and ASV27/28 could be two different species of *Nitrosopumilus*. I would be curious to know how similar ASV1 and ASV27/28 are to the *Nitrosopumilus* found in the microbiome of the black coral *Antipathes ceylonensis* (Liu et

al. <https://academic.oup.com/femsle/article-abstract/365/15/fny167/5047306>).

- Lines 273-276: Victorgorgiidae and Keratoisididae are represented by $n = 1$ and Paramurcia is represented by $n = 2$, so I would be cautious here and specifically highlight the biological replication limitation in this paragraph.
- Lines 273-276: Also note, you may be able to compare main Keratoisididae ASV against clones from Penn et al. paper which also sampled several bamboo corals from Gulf of Alaska seamounts. <https://journals.asm.org/doi/full/10.1128/aem.72.2.1680-1683.2006>
- Lines 285-287: The other dominant microbiome member found in corals is Spirochaetes, see van de Water et al. <https://www.nature.com/articles/srep27277> and Holm & Heidelberg <https://www.frontiersin.org/journals/microbiology/articles/10.3389/fmicb.2016.00917/full>

Reviewer #1 (Comments for the Author):

The authors conduct a 16S metabarcoding survey of several deep-sea corals, water, and sediment from the Magellan Seamounts in the western Pacific to characterize those corals' microbiomes. This study focuses on under-studied coral taxa including black corals and several octocoral families and identifies *Nitrosopumilus* as an abundant member of several black coral microbiomes and detects general differences between coral taxa at multiple taxonomic levels. The methods seem appropriate. I suggest that the paper be accepted with minor revisions.

My main criticism is a lack of important details in the methods which I list below and the interpretation of the results. The authors propose that black corals associate with *Nitrosopumilus* because zooplankton comprise a larger proportion of their diet compared to other coral taxa. The literature cited does not support the claim that black corals rely on more zooplankton. Several of the cited works do not even refer to the topic of the sentences they are associated with. I don't know if a citation tool messed up or the works were not thoroughly read. The authors also imply that generally octocorals eat primarily phytoplankton which is not true. In general, I think more uncertainty should be expressed. Something like black corals MAY rely on less phytoplankton and this MAY be the basis of their association with *Nitrosopumilus*. Otherwise, the rest of the discussion was good and the authors handled the other abundant microbes well.

Response: We have addressed the reviewer's concerns by responding to the specific comments listed below.

Line 28: remineralization of zooplankton? There must be a more direct way to communicate this. You mean the nitrogen waste. Is there evidence that schizopathids prefer zooplankton more so than the other deep-sea corals?

Response: We have replaced "remineralization" with "catabolism" for clarity. We introduced some previous studies (Line 66-69) which suggest that black corals may rely more exclusively on zooplankton than octocorals.

Line 30: cladopathid corals "hosted" ...

Line 39: Microbiomes

Line 50: disagree with likely. Should say MAY be attributed to

Response: Updated as suggested.

Line 67: It is not accurate to generalize octocorals as only consuming phytoplankton.

Ref 11 shows that some shallow water octos in the red sea eat exclusively phytoplankton.

This is not very informative for deep-sea corals where there isn't light

Ref 12 doesn't even mention octocorals at all.

Ref 13 shows *P. clavata* can capture things other than zooplankton.

There are several studies on deep-sea corals diets. I think you should reframe this by saying black corals may rely more exclusively on zooplankton than octocorals.

Response: Ref 12 has been removed and the suggested phrase has been added to the end of the sentence.

The stable isotope results in Ref 16 suggest that *Paramuricea macrospina* primarily eats zooplankton and that *Leiopathes* has a lower contribution of zooplankton in its diet. This runs counter to the generalizations proposed by the authors.

Response: Although we had also noted this point, we included this exception in the introduction to provide a more comprehensive overview on the diversity of the ecological and nutritional niches (as reflected by the isotopic analysis) of black corals.

Line 96: Missing citation from Dannenberg. This was published as Vohsen et al 2020. Microbiome. Deep-sea corals provide new insight into the ecology, evolution, and the role of plastids in widespread apicomplexan symbionts of anthozoans.

Response: We thank the reviewer for bringing this work to our attention. However, the suggested paper investigates eukaryotic apicomplexan symbionts associated with anthozoans (including black corals), whereas our study focuses on the coral microbiome, specifically bacteria and archaea, and is therefore not directly relevant to our research.

Line 162: What were the parameters used for all these tools?

Response: Program parameters have now been provided.

Line 167: you mean if more than 1 of the top 10 hits are eukaryotes? This is an unusual criterion. Why was this chosen?

Response: We selected this stringent cutoff to remove any sequences that may have originated from eukaryotes.

Line 186: were the permanova analyses conducted on ASVs or higher taxonomic level? What distance metric was used? Water and sediment should have been excluded from order, family, and genus comparisons. Is this how they were conducted?

Response: PERMANOVA analyses were performed on data matrices in which microbial taxonomy was merged at genus level. The Bray-Curtis distance metric was used to calculate the distances for these analyses. This information has now been added to the Materials & Methods section.

All details regarding the stats analyses are missing from the methods

Response: The details of the statistical analyses have been added into the Materials & Methods section.

Table 1 should state the species for each of those samples

Response: The lowest level of taxonomic classification determined for the corresponding coral samples has been added to Table 1.

Line 212: Ref 29 does not conclude that *Nitrosopumilus*' chemoautotrophy occurs in other deep-sea corals. It discusses chemoautotrophy by sulfur oxidizing SUP05 based on 16S studies and not metagenomics. Reword this to be clear.

Response: We thank the reviewer for pointing this out, reference 29 has been removed from this context to ensure accuracy.

Line 223: Were the permanova analyses also conducted on genus level? What distance metric was used for NMDS and permanova? Do you mean ASVs that could not be assigned to genus were excluded?

Response: Yes, both NMDS and PERMANOVA analyses were performed on data matrices, in which microbial taxonomy was merged at the genus level. The key difference is that, water and sediment samples were excluded from the PERMANOVA analyses. In both analyses, the Bray-Curtis distance metric was used to calculate the distances. Coral samples that could not be assigned to a genus were excluded, as is now clearly stated in the legend of Figure 3.

Line 225: that figure should also give the lowest taxonomic id for each sample

Response: The genus-level classification of the coral samples has now been added to Figure 4.

Line 196: B01 does not say *Bathypathes* in the figure below

Response: JL311_B01 is indeed labelled as *Bathypathes* in Figure 5. The reviewer may have instead been referring to other non-*Bathypathes* B01 samples, such as JL317_B01, JL313_B01, and JL316_B01.

Paramuricea is in the Paramuriceidae and not the Plexauridae

Response: The assignment of the genus *Paramuricea* (JL303_B05 and JL303_B26) to the family Plexauridae is likely introduced by errors in the NCBI taxonomy database (<https://www.ncbi.nlm.nih.gov/Taxonomy/Browser/wwwtax.cgi?id=44197>), as reflected by the taxonomy of the GenBank records listed below (there might be more). In our manuscript, we have manually reassigned the genus *Paramuricea* to the family Paramuriceidae. The statistical analysis and NMDS plot have also been updated to reflect this reassignment.

PP073611.1 <https://www.ncbi.nlm.nih.gov/nuccore/PP073611.1>

PP410649.1 <https://www.ncbi.nlm.nih.gov/nuccore/PP410649.1>

PP410648.1 <https://www.ncbi.nlm.nih.gov/nuccore/PP410648.1>

PP410650.1 <https://www.ncbi.nlm.nih.gov/nuccore/PP410650.1>

PP073707.1 <https://www.ncbi.nlm.nih.gov/nuccore/PP073707.1>

Figure 5: those "genus" classifications are often just the closest match in GTDBTK database. Provide a higher taxonomic level so they are interpretable

Response: The higher taxonomic levels (from class to genus) of the ASV classifications have been added to the Figure 5.

No mention of genome skims in methods. No mention how they got coral barcodes from skims.

Response: Details on the sequencing and processing (QC and assembly) of the genome-skimming data, as well as the identification of barcoding sequences, are now described in the

sections “DNA Extraction, Library Construction and Sequencing” and “Host Classification and Phylogeny Inference”.

No mention of genome skims in data availability or are they some of those SRAs?

Response: Genome skimming data have been deposited in the NCBI SRA, with accession numbers listed in the “Data Availability” section.

Was there anything about the bathypathes that could explain why 2 host different nitrosopumilus? Maybe those bathys were from different sites or different depths. Authors should mention that nitrosopumilus is a very common member of the bacterioplankton.

Response: Based on the metadata (table below), neither sampling depth nor location could explain why JL305_B10 and JL311_B01 harbour two different Nitrosopumilus ASVs. The ubiquitous distribution of Nitrosopumilus across marine and terrestrial environments has now been emphasized in its introduction.

Coral genus	Sample	Nitrosopumilus	Depth	Latitude	Longitude
	JL303_B16	ASV1	1428.5	23.858999	148.749837
	JL303_B29	ASV1	1428.5	23.858999	148.749837
Bathypathes	JL304_B04	ASV1	3101	23.524038	148.576579
	JL311_B01	ASV27	1252	16.999134	154.273926
	JL305_B10	ASV28	1678	20.481838	151.140934
Parantipathes	JL305_B08	ASV1	1671	20.481698	151.141113

Reviewer #2 (Comments for the Author):

This study uses 16S amplicon surveys of the V4-variable region to examine the bacterial and archaeal communities of 30 deep-sea coral samples, belonging to 9 different coral families. The main limitation of this study is that the specimen collections contain few biological replicates, with many genera or even families at n =1 or n =2. The authors have done their best to be able to say something meaningful by combining the data at the family level. The results

reflect several expected findings plus some bits of new information. For example, it is in no way unexpected based on existing literature to find that the microbial families are different between black corals and octocorals, different between coral families, and different between coral genera. It is similarly not unexpected to find Endozoicomonas present in some of the deep-sea coral-associated microbial communities. What is new information is the high relative abundance of archaea, specifically Nitrosopumilus in the microbiomes of the black coral family Schizopathidae, but not in family Clathrotopathidae nor any of the octocoral families. The study also provides new information in the form of specific ASVs that are dominant in coral families and genera that have not yet been characterized, but with sample sizes of 1, 2, or even 4, it is of less value since it is unclear how generalizable these data might be to other geographic locations or even other coral individuals. The introduction and results/discussion sections contain appropriate references and place the study in proper context of the existing literature.

Response: We have addressed the reviewer's concerns by responding to the specific comments listed below.

Specific comments:

(Note that the line numbers are slightly different between the Word docx version and the PDF version; these numbers refer to the docx version of the manuscript)

Response: No need to respond.

Abstract:

- Please specify in the abstract the variable region targeted for 16S amplicon surveys. E.g., "In the present study, we used 16S rRNA gene amplicon sequencing targeting the V4 variable region to characterize the microbiomes..."

Response: Updated as suggested.

Introduction:

- Line 67: The sentence cites [11-13] about octocorals, but reference 12 is about black coral *Antipathella wollastoni*.

Response: Sorry for the confusion, reference 12 has now been removed.

- Lines 86-88: Please consider including Gray et al. (<https://academic.oup.com/femsec/article/76/1/109/520994>) for octocorals and Chapron et al. (<https://www.frontiersin.org/journals/microbiology/articles/10.3389/fmicb.2020.00275/full>) for scleractinians.

Response: The suggested references have been included in the manuscript.

- Lines 95-97: While it is absolutely true that deep-sea black coral microbiomes are extremely understudied, this sentence could be more accurate. Citation [34] references a master's thesis but there is also one peer-reviewed example (Penn et al., <https://journals.asm.org/doi/full/10.1128/aem.72.2.1680-1683.2006>) albeit not a very helpful one since the scientific name of the black coral sampled is not provided and it from early days of clone libraries. However, the coral is from a seamount in the Gulf of Alaska, so worth making the connection.

Response: The work described in the suggested reference has now been incorporated into the manuscript.

Methods:

- Lines 118-120: Please specify whether each coral was in a separate compartment of the collection box, or whether multiple specimens were in the same container (in physical contact with each other). It allows readers to gauge potential for cross-contamination.

Response: Multiple specimens from the same dive were stored in the same biological sample box. As pointed out by the reviewer, rinsing specimens three times with sterile filtered seawater definitely helps reduce contact-contamination. Such information has been added into the Sample collection and processing section.

- Lines 120-121: How were corals handled during photography? E.g., were sterile forceps/gloved hands used to transfer them from the submersible biobox to sterile trays? Otherwise, be clear on this possible route for contamination.

Response: During specimen photography, onboard scientists balanced steps to reduce contamination to the specimens and minimize time used in the processing which could affect microbiome. Therefore, they processed specimens by wearing sterile gloves and cleaning trays with running fresh water between handling different specimens. While this step was not processed under germ-free conditions, subsequent subsampling in the BSC followed strict germ-free protocols, and rinsing samples three times with sterilized seawater helped remove loosely attached microorganisms introduced during sampling. These details have now been added into the “Sample collection and processing” section.

- Lines 121-123: Specimens rinsed 3x with sterile filtered seawater-good, this definitely helps reduce contact-contamination!

Response: No need to respond.

- Line 125: Why are there 2 preservation methods (liquid nitrogen and ethanol)? Were all corals preserved both ways for safety, or were some preserved one way and some the other? Preservation method can affect the diversity detected in the microbiome, so if the sequenced samples are of mixed preservation provenance, you should indicate which samples were preserved which ways, since it could be the cause of some of the patterns you are seeing.

Response: Coral samples in this study were preserved using two methods—freezing and storage in 100% ethanol—for sequencing purposes, with absolute ethanol also supporting improved observation. Due to the small biomass of some specimens, ethanol-preserved samples were used when frozen samples ran out. Previous research has shown that preserving fauna samples in 95-100% ethanol allows for rapid penetration of cellular membranes and effective deactivation of DNases [1]. Additionally, it has been suggested that ethanol at concentrations of 95% or higher is a viable option for long-term storage of fecal samples for microbiome analyses [2]. According to the preservation details in the form attached below corresponding to the order of samples shown in Figure 5, the samples were either all from the same family (e.g., JL307_B07, JL305_B13, and JL304_B13 from Cladopathidae) or included both ethanol- and frozen-preserved specimens from the same family (e.g., JL305_B10 and JL303_B16 from Shizopathidae, as well as JL307_B09 and JL300_B05 from Primnoidae). As shown in Figure 5, Microbiome profiles were consistent across preservation methods within these groups, therefore, the preservation method should have minor effect on the community structure results.

References:

- [1] King JR and Porter SD. 2004. Recommendations on the use of alcohols for preservation of ant specimens (Hymenoptera, Formicidae). *Insectes Soc*, 51:197–202.
- [2] Song SJ, Amir A, Metcalf JL, Amato KR, Xu ZZ, Humphrey G, Knight R. 2016. Preservation methods differ in fecal microbiome stability, affecting suitability for field studies. *mSystems*, 1(3):10-1128.

Sample ID	Preservation method
JL316_B11	Stored in freezer at -80°C
JL307_B09	Absolute ethanol, stored in refrigerator at 4°C
JL307_B20	Stored in freezer at -80°C
JL317_B09	Stored in freezer at -80°C
JL301_B08	Stored in freezer at -80°C
JL304_B07	Stored in freezer at -80°C
JL313_B05	Stored in freezer at -80°C
JL300_B21	Stored in freezer at -80°C
JL300_B05	Absolute ethanol, stored in refrigerator at 4°C
JL316_B01	Stored in freezer at -80°C
JL303_B27	Stored in freezer at -80°C
JL306_B04	Stored in freezer at -80°C
JL316_B20	Stored in freezer at -80°C
JL313_B01	Stored in freezer at -80°C
JL300_B04	Stored in freezer at -80°C
JL303_B26	Stored in freezer at -80°C
JL303_B05	Stored in freezer at -80°C
JL312_B19	Stored in freezer at -80°C
JL307_B07	Absolute ethanol, stored in refrigerator at 4°C
JL305_B13	Absolute ethanol, stored in refrigerator at 4°C
JL304_B13	Absolute ethanol, stored in refrigerator at 4°C
JL317_B01	Stored in freezer at -80°C
JL307_B11	Stored in freezer at -80°C
JL305_B08	Stored in freezer at -80°C
JL317_B03	Stored in freezer at -80°C
JL304_B04	Stored in freezer at -80°C
JL303_B29	Stored in freezer at -80°C
JL303_B16	Absolute ethanol, stored in refrigerator at 4°C
JL305_B10	Absolute ethanol, stored in refrigerator at 4°C
JL311_B01	Stored in freezer at -80°C

• Figure 1 legend: Panel M is listed between G and H, when it should be at the end.

Response: Updated as suggested.

• Lines 152-153: Please include citations for primers. E.g., the citation for 515F/806R is Caporaso et al. 2011 <https://www.pnas.org/doi/abs/10.1073/pnas.1000080107>.

Response: Updated as suggested.

• Lines 150-156: Please describe the methods used to assess DNA quality; e.g., spectrophotometrically like NanoDrop, fluorometrically like Qubit.

Response: Details of the DNA quality assessment have been added to the manuscript.

- Lines 150-156: Please describe any positive controls (e.g., mock community) and negative controls (e.g., kit extraction blanks, PCR negatives, sequencing no-template-controls) that were used.

Response: The use of sterilized water as the negative control in PCR amplification has been added to the manuscript.

- Lines 173-177: Please provide reference(s) for the primers listed.

Response: References for the primer pairs C2/D2 and dgLCO1490/dgLCO2198 have been provided.

- Good job describing the bioinformatics, including citations for tools and version numbers for databases. Noting that the raw data have been made available in NCBI and the supplemental files.

Response: No need to respond.

Results and Discussion:

- Table 1: It would be most helpful to include genera names, when possible, for samples in addition to the sample ID numbers.

Response: Updated as suggested.

- Line 198: Given that ASV27 and ASV28 differ by one base pair, they are likely the same organism and that is a sequencing artifact. I agree that ASV1 and ASV27/28 could be two different species of *Nitrosopumilus*. I would be curious to know how similar ASV1 and ASV27/28 are to the *Nitrosopumilus* found in the microbiome of the black coral *Antipathes ceylonensis* (Liu et al. <https://academic.oup.com/femsle/article-abstract/365/15/fny167/5047306>).

Response: Although the comment is not directly relevant here, I nonetheless did some comparisons. The similarities of ASV1, ASV27, and ASV28 to *Nitrosopumilus* detected in *Antipathes ceylonensis* (accession SRR5931567) are 97.414, 99.138, and 98.707%, respectively.

- Lines 273-276: *Victorgorgiidae* and *Keratoisididae* are represented by $n = 1$ and *Paramurcia* is represented by $n = 2$, so I would be cautious here and specifically highlight the biological replication limitation in this paragraph.

Response: We agree with the reviewer's concern. Since these are all the samples available to us, we included them in the analysis despite the lack of biological replicates. Nonetheless, we have now highlighted in the manuscript the limitations of the conclusions that can be drawn from above families given the insufficient number of samples.

- Lines 273-276: Also note, you may be able to compare main *Keratoisididae* ASV against clones from Penn et al. paper which also sampled several bamboo corals from Gulf of Alaska seamounts. <https://journals.asm.org/doi/full/10.1128/aem.72.2.1680-1683.2006>

Response: The abundance of Alphaproteobacteria, Gammaproteobacteria, Firmicutes, Bacteroidetes, and Acidobacteria reported in bamboo corals from the Gulf of Alaska

seamounts has now been introduced and compared with the microbial community observed in our bamboo coral.

- Lines 285-287: The other dominant microbiome member found in corals is Spirochaetes, see van de Water et al. <https://www.nature.com/articles/srep27277> and Holm & Heidelberg <https://www.frontiersin.org/journals/microbiology/articles/10.3389/fmicb.2016.00917/full>

Response: We thank the reviewer for pointing this out. Spirochaete has now been introduced as one of the microbial taxa that have previously been identified as dominant species in shallow-water corals.

Re: Spectrum02093-25R1 (**Distinct Patterns of Microbial Association Between Deep-sea Corals from the Western Pacific Magellan Seamounts**)

Dear Prof. Pei-Yuan Qian:

Your manuscript has been accepted, and I am forwarding it to the ASM production staff for publication. Your paper will first be checked to make sure all elements meet the technical requirements. ASM staff will contact you if anything needs to be revised before copyediting and production can begin. Otherwise, you will be notified when your proofs are ready to be viewed.

Sincerely,
Konstantinos Kormas
Editor
Microbiology Spectrum

Reviewer #2 (Comments for the Author):

The authors have updated the manuscript and responded appropriately to reviewer suggestions.

In Methods, lines 175-178, the text now lists references [20] and [21] from the supplemental file but as written, readers will assume they are references 20 and 21 from the bibliography. Need to move references for Trimmomatic and SPAdes to the main bibliography.